# Triglycerides, Obesity and Education Status Are Associated with the Risk of Developing Type 2 Diabetes in Young Adults, Cohort Study

**DOI:** 10.3390/jpm13091403

**Published:** 2023-09-19

**Authors:** Evgeniia V. Garbuzova, Lilia V. Shcherbakova, Oksana D. Rymar, Alyona D. Khudiakova, Victoria S. Shramko, Yulia I. Ragino

**Affiliations:** Research Institute of Internal and Preventive Medicine—Branch of the Institute of Cytology and Genetics, Siberian Branch of Russian Academy of Sciences (IIPM—Branch of IC&G SB RAS), 630089 Novosibirsk, Russia; stryukova.j@mail.ru (E.V.G.); 9584792@mail.ru (L.V.S.); orymar23@gmail.com (O.D.R.); alene.elene@gmail.com (A.D.K.); nosova@211.ru (V.S.S.)

**Keywords:** cohort study, persons 25–44 years old, type 2 diabetes mellitus, obesity, hypertriglyceridemia, educational status

## Abstract

Background: It is important to determine the influence of traditional risk factors on the development of type 2 diabetes mellitus (T2DM) in young adults. Goal of the research: To study the incidence of T2DM and factors that increase the risk of its occurrence during the observation of a cohort of young adults. Materials and methods: 1341 people aged 25–44 were included in the study from 2013 to 2017, of whom 622 were men (46.4%). The examination included anamnesis, anthropometric data, and a blood test. Cases of developed T2DM were identified by comparing the Diabetes Mellitus Register, medical records of patients, and the database of examined individuals from 2019 to 2023. T2DM Results: In the examined population, 11 participants (0.82%) developed T2DM. The prevalence of T2DM was 0.96% in men and 0.69% in women. Patients with T2DM had a higher waist circumference, BMI, SBP, TG, and lower HDL than patients without T2DM, and were also less likely to have a higher education. The risk of developing T2DM increases 6.5 times at a BMI of ≥30 kg/m^2^, and 5.2 times at a TG level of ≥1.7 mmol/L, regardless of other risk factors. In the absence of a higher education, the risk of developing T2DM is increased by 5.6 times. Conclusion: In young people, high triglyceride levels, obesity, and a low level of education are associated with the risk of developing type 2 diabetes, regardless of other factors.

## 1. Introduction

Type 2 diabetes mellitus (T2DM) is a disease with a rapidly growing prevalence worldwide that is largely associated with obesity and a sedentary lifestyle [1]. The prevalence of established T2DM in Russia is 3.2% [2]. At the age of 25–44 years, according to the NATION study, the prevalence of T2DM is 1.5%. By age subgroups, the prevalence of T2DM is: at the age of 25–29 years, 1.14%; at the age of 30–34 years, 0.82%; at the age of 35–39 years, 1.64%; and at the age of 40–44 years, 2.75% [3]. At the same time, the proportion of participants with prediabetes and T2DM increased with increasing body mass index (BMI).

The global trend of increasing T2DM prevalence, including among all patients with DM [4], is also confirmed by data from Russian studies [5,6]. Global and Russian trends also predict an epidemic of type 2 diabetes among people of young working age. The influence of lifestyle and weight loss on the risk of developing T2DM may vary for different people depending on age and body composition [7]. At the same time, given the increasing prevalence of obesity in the Russian Federation at a young age [3,8], it becomes important to determine its influence as well as the influence of other traditional risk factors on the development of T2DM in young adults for personalized prevention.

The aim of this study was to study the incidence of type 2 diabetes mellitus and factors that increase the risk of its occurrence during the observation of a cohort of young adults aged 25–44 years.

## 2. Materials and Methods

The study design was a prospective cohort study. Based on the IIPM—Branch of IC&G SB RAS, a survey of the population of Novosibirsk from 2013 to 2017 was conducted. The study was approved by the local ethics committee of IIPM—Branch of IC&G SB RAS (Proto-col No. 6/2013 of 25 June 2013).

The Territorial Compulsory Health Insurance Fund’s base, which consists of residents of one of Novosibirsk’s districts between the ages of 25 and 44, was utilized to create the sample. Using a random number generator, a sample of 2500 people was selected at random to reflect the population. Since invitations to young age groups are the least likely to be accepted, step-by-step incentive approaches, such as invites sent via mail, phone calls, and media messages were employed. At the screening, 1512 individuals were assessed; those with any known form of diabetes, women who were pregnant, and individuals who had fatal events occur during follow-up were removed. 1341 individuals were included in the final analysis, including 622 men (46.4%) and 719 women (53.6%) (Figure 1). The respondents’ average age was 37.08 [31.75; 41.75]. Informed consent of all was obtained for the examination and processing of personal data.

The endpoint in the form of cases of developed T2DM were identified by comparing the Diabetes Mellitus Register, medical records of patients, and the database of examined individuals from 2019 to 2023. A group of medical professionals with training in standardized epidemiological screening techniques performed the screening. The survey program contained a number of components, such as demographic and social data, a smoking habits survey, a socioeconomic survey, Rose’s cardiological questionnaire, anthropometry, 2-fold blood pressure measurement, spirometry, ECG recording with transcription in accordance with the Minnesota Code, and others.

Educational level was divided into 3 groups: higher education, secondary and specialized secondary education, and primary and incomplete secondary education. All groups were included in the analysis, and the final analysis included higher education against all other types of education. Waist circumference (WC) was measured with a centimeter tape placed horizontally in the center between the sacral iliac bone and the lower edge of the costal arch [9,10]. Body mass (kg) divided by the square of height (m^2^) is the formula used to calculate the body mass index (BMI) [10]. Elevated BMI was considered >30 kg/m^2^. Physical activity was considered sufficient if more than 3 h of physical activity per week [10].

After a 5 min break, blood pressure was checked twice with a 2 min gap on the right hand while seated and registered as the average of the two readings using an Omron M5-I automated tonometer. Systolic blood pressure (SBP) of more than 140 mmHg and/or diastolic blood pressure (DBP) of more than 90 mmHg were classified as arterial hypertension (AH) [10].

Smokers were defined as those who smoked at least one cigarette each day.

On an empty stomach, 12 h after eating, a single blood sample was taken from the ulnar vein. On a KoneLab 30i automatic biochemical analyzer (Finland), blood parameters for lipid profile (total cholesterol (TCH), low-density lipoprotein cholesterol (HDL-C), triglycerides (TG), lilow-density lipoprotein cholesterol (LDL-C)), glucose, albumin, urea, and creatinine were analyzed using an enzyme-based approach. The formula used to convert serum glucose into plasma glucose was plasma glucose (mmol/L) = −0.137 + 1.047 × serum glucose (mmol/L) [10]. The levels of low-density lipoprotein cholesterol were calculated using the Friedwald formula (TCH − (TG/5) − HDL-C). Decreased blood levels of HDL-C were estimated to be less than 1 mmol/L for men and less than 1.2 mmol/L for women, while elevated blood levels of TG were estimated to be more than 1.7 mmol/L, and elevated blood levels of LDL-C were estimated to be 3 mmol/L [10]. The calculation of GFR was carried out according to the formula CKD-EPI (Chronic Kidney Disease Epidemiology Collaboration), taking into account race, gender, age, and serum creatinine level.

The SPSS software tool (version 13.0) was used to statistically process the results. The Kolmogorov-Smirnov criterion was used to examine the distribution. Continuous variables have non-normal distributions; hence, the data are presented as Me [25; 75], where Me is the median and 25 and 75 are the first and third quartiles, respectively. For categorical indicators, the data are presented as absolute and relative values, *n* (%). To compare two independent samples, the nonparametric Mann-Whitney U-test was employed. To compare the fractions, Pearson’s chi-squared test was employed. Multivariate models of logistic regression analysis were performed to identify independent prognostic predictors of the development of diabetes mellitus. The critical significance level of the null hypothesis (p) was assumed to be 0.05.

The work was supported by the grant of the Russian Science Foundation No. 21-15-00022 (statistical processing, collection of endpoints). The funding organization played no role in the development of the study, data collection, analysis, interpretation of data or writing of the manuscript.

## 3. Results

In the examined population, 11 cases (0.82%) of T2DM developed during the cohort observation. The prevalence of T2DM in men was 0.96%, and in women it was 0.69%. During the first stage of our study, the clinical and anamnestic data of patients with and without T2DM were analyzed. Patients with developed diabetes mellitus, compared to patients without T2DM, had a waist circumference 1.2 times higher (101.00 (90.00; 122.00) vs. 85.00 (76.0; 95.40]), *p* < 0.001), a BMI 1.3 times higher (32.37 (29.78; 39.44) vs. 25.01 (22.04; 28.70), *p* < 0.001), an SBP 1.1 times higher (132.50 (121.50; 140.00) vs. 119.00 (110.00; 129.00), *p* = 0.033), a TG 1.9 times higher (1.75 (1.22; 2.89) vs. 0.94 (0.68; 1.38), *p* = 0.003), and an HDL-C 1.3 times lower (1.03 (10.90; 1.16) vs. 1.29 (1.08; 1.52). *p* = 0.001), and were also less likely to have a higher education. There were no differences in age, sex, smoking status, level of physical activity, civil partnership status, employment, DBP, presence of AH, LDL-C, TCH, Creatinine, glomerular filtration rate (GFR), Albumin, Urea, and Glucose levels between patients with and without T2DM. The characteristics of the examined patients are presented in Table 1.

To study the associations of cardiometabolic parameters with the risk of developing diabetes mellitus, a logistic regression analysis was performed (Table 2). The analysis showed that the risk of developing diabetes mellitus is associated with an increase in SBP (by 35.7% per 10 mm Hg), TG (by 47.5% per 1 mmol/L and by 9 times at a level of TG greater than 1.7 mmol/L), WC (by 8.6% per 1 cm and by 13 times in the presence of abdominal obesity (AO)), BMI (by 20% per 1 kg/m^2^), and a decrease in HDL-C (by 45 times per 1 mmol/L and by 4.4 times at a level of HDL-C less than 1.1 mmol/L for men and <1.2 mmol/L for women), as well as with other types of education, except higher education.

The next stage was the construction of models for multivariate logistic regression analysis (Table 3). Model 1 included gender, age, TG level, and the presence of AO (determined by WC); in Model 2, the presence of AO was replaced by BMI. Models 3 and 4 additionally include education status.

The multivariate logistic regression analysis showed that TG, BMI, and education status were associated with the risk of developing T2DM. In Model 4, the risk of developing T2DM increases 6.5 times at a BMI of ≥30 kg/m^2^, as well as 5.2 times at a TG level of ≥1.7 mmol/L, regardless of other cardiometabolic risk factors. Additionally, in the absence of a higher education, the risk of developing T2DM increases 5.6 times.

## 4. Discussion

The main purpose of this study was to study the influence of various cardiometabolic and social factors on the risk of developing T2DM. After conducting a study on a sample of young participants in Novosibirsk, we found that high triglyceride levels, a high body mass index, and a low level of education were associated with the risk of developing T2DM, independent of other cardiometabolic risk factors. In general, our data are consistent with previous studies. The prevalence of T2DM in young adults in Novosibirsk was studied earlier and amounted to 2.2% [11], which is consistent with the all-Russian data of the NATION study. According to the results of this study, 0.82% of people in the study developed T2DM (taking into account the fact that patients with a history of T2DM were excluded from the analysis).

Chronic diseases like diabetes and obesity are becoming more prevalent everywhere [12]. Body mass index, as well as diabetes and insulin resistance are closely related. NEFA, glycerol, hormones, cytokines, proinflammatory chemicals, and other compounds that are involved in the development of insulin resistance are present in higher amounts in obese people. Insulin resistance with impairment of β-cell function leads to the development of diabetes [13]. According to the NATION study, the number of participants with prediabetes and T2DM increased as BMI increased. In the group with a BMI of less than 25 kg/m^2^, the prevalence of T2DM was 1.1%. In the group with a BMI of more than 25 and less than 30 kg/m^2^, the prevalence of T2DM was 3.9%, and among obese people, the prevalence of T2DM was 12.0%. The data obtained show that T2DM is more common in obese people than in people with a normal body mass (*p* < 0.001) [3]. In China, in a cross-sectional study of 5860 people, the chance of having T2DM in people with abdominal obesity was 1.55 times higher (95% CI 1.08–2.24) [14]. At the same time, in studies that included high-risk overweight and obese populations, weight loss was an important factor in preventing diabetes or delaying its development; a greater benefit was usually observed with greater weight loss [15,16]. In a cohort study of individuals aged 56.1 years (range 50–65), obesity and an unfavorable lifestyle were associated with a higher risk of developing T2DM, regardless of genetic predisposition. Even among people with a healthy lifestyle, obesity was associated with a more than 8-fold risk of developing T2DM compared to people with normal weight in the same lifestyle group [17]. In the study, all patients with T2DM had a BMI of more than 25 kg/m^2^, while a BMI of more than 30 kg/m^2^ is associated with a 6-fold risk of developing T2DM in young patients, which underscores the importance of early prevention in the form of recommendations for weight loss at any age. BMI does not accurately reflect the adipose tissue distribution throughout the body, and it is unable to distinguish between fat mass and muscle mass. Visceral adipose tissues seem to play a significant role in the emergence of metabolic problems linked to obesity [18,19]. The distribution of adipose tissue is a well-documented and major factor linked to the development of insulin resistance, and WC, which is an index of body fat distribution, could be used as a criterion for evaluating abdominal obesity [20]. Along with BMI, WC can be used to assess the distribution of adipose tissue in young adults. In our study, an increase in WC was associated with an increase in the risk of 2DM in a single-factor analysis; however, BMI turned out to be more significant in multivariate analysis models. Our result was consistent with some earlier studies which revealed that the TG level might be an influential factor for T2DM in young men [21] and women [22]. The results of an 8-year prospective observation of the middle-aged and elderly Chinese population (61.49 ± 13.85 at the time of inclusion) showed that triglycerides are an independent predictor of the development of T2DM. Compared with the group with a normal level of TG, the risk coefficients for developing T2DM in the borderline (TG 1.7–2.25 mmol/L) and high (TG ≥ 2.26 mmol/L) TG groups were 1.30 (95% CI 1.04–1.62) and 1.54 (95% CI 1.24–1.90), respectively. Survival analysis has shown that higher levels of TG can predict an earlier onset of T2DM [23]. In another Chinese study of 7329 people over 45 years of age, hypertriglyceridemia also increased the risk of developing T2DM by 1.9 times (95% CI 1.49–2.46) [24]. In a prospective 5-year follow-up of 5085 younger patients without T2DM, 42.8 ± 9.0 years old, every 10 mg/dL increase in triglyceride levels significantly increased the risk of developing T2DM by 4%. This relationship persisted even when the triglyceride level remained within the generally accepted norm (<150 mg/dL (<1.7 mmol/L), *p* < 0.001) [25]. In our study, the risk of T2DM was 5.2 times higher in patients with a TG level of more than 1.7 mmol/L in young adults. T2DM can be caused by a variety of things, both hereditary and acquired, but the exact mechanism is yet unknown. As an endocrine organ, adipose tissue can affect the metabolism of glucose and lipids. TGs are the most prevalent lipid in adipose tissue [26]. The free fatty acid (FFA) metabolic pathways can be mentioned as the probable mechanism connecting TG to T2DM. Because adipose tissue with an excessive amount of FFA, resistin, TNF-α, interleukin-6, and other substances might result in insulin resistance [27]. Furthermore, raising the level of FFA produced by TGs can increase insulin resistance [28].

The level of education was also previously studied in the context of the risk of developing T2DM: 17.2% of the cases of diabetes in men and 20.1% of the cases in women were associated with a lower level of education in Sweden in all age groups; diabetes mellitus was more common in the group with a low level of education in groups older than 70 years [29]. In a population of 53,159 Danish men and women aged 50–64 years with a follow-up period of 14.7 years, differences in the incidence of T2DM were estimated depending on the level of education per 100,000 person-years. Compared with a high level of education, a low level of education was associated with 454 (95% CI 398–510) additional cases of T2DM, and an average level of education was associated with 316 (CI 268–363) additional cases [30]. At the same time, the authors believe that different susceptibilities to being overweight or having obesity are important mechanisms in the relationship between education and the incidence of T2DM. Previous studies have shown that from 8% [31] to 64% [32] of the relationship between the level of education and type 2 diabetes can be explained by different exposures to being overweight or having obesity [33]. In another Danish study, which included 83,759 and 91,083 participants aged 30–65 years, from a cohort of patients with T2DM and cardiovascular diseases, respectively, a low level of education was positively associated with a higher risk of developing T2DM (OR 1.24, 95% CI 1.04–1.48) [34]. Education establishes an individual’s non-material resources, such as knowledge, skills, and self-efficacy, which lowers obstacles and enables people to be more receptive to health messages and convert those messages into healthy behaviors. Additionally, people with a lower level of education are less physically active, smoke more, and consume fewer fruits and vegetables than people with a high level of education [30], which may have an indirect effect on the development of T2DM, but was not separately evaluated in this study. The analysis of our data provides very important additional data on the risk factors for developing T2DM in a young population. The results show that despite the fact that people with obesity and dyslipidemia may benefit from lifestyle changes, their high risk of developing T2DM may persist depending on their initial level of education.

Gender and age in the conducted study did not affect the risk of developing T2DM; perhaps this is due to the initially young age of the population and the insufficient follow-up period for these variables to have an impact.

This study’s strengths include the large sample size and extended follow-up. We were able to account for numerous potential confounders because of the inclusion of numerous comprehensive clinical and biochemical examinations. After reading various references, our research is one of the few in Russia and the first in Novosibirsk evaluating risk factors for the development of T2DM in young adults.

This study has several limitations. First, this study was a single-center investigation, and the enrolled participants were primarily residing in the city. Further investigations should be conducted as multi-center studies that include various areas. Second, limitations of the study include lack of access to information regarding family history of diabetes mellitus, age of diagnosis, and lack of information about gestational diabetes. There was also no information about glucose levels during the observation period, which does not allow to exclude latent T2DM. Also, the estimate of the effect of educational level on type 2 diabetes may be inflated due to unmeasured confounding factors from early life, such as family socioeconomic position, family history of T2DM and fetal/neonatal factors.

## 5. Conclusions

In young people, high triglyceride levels, obesity, and a low level of education are associated with the risk of developing type 2 diabetes, regardless of other factors. These data can be taken into account when developing personalized preventive measures for different age groups.

## Figures and Tables

**Figure 1 jpm-13-01403-f001:**
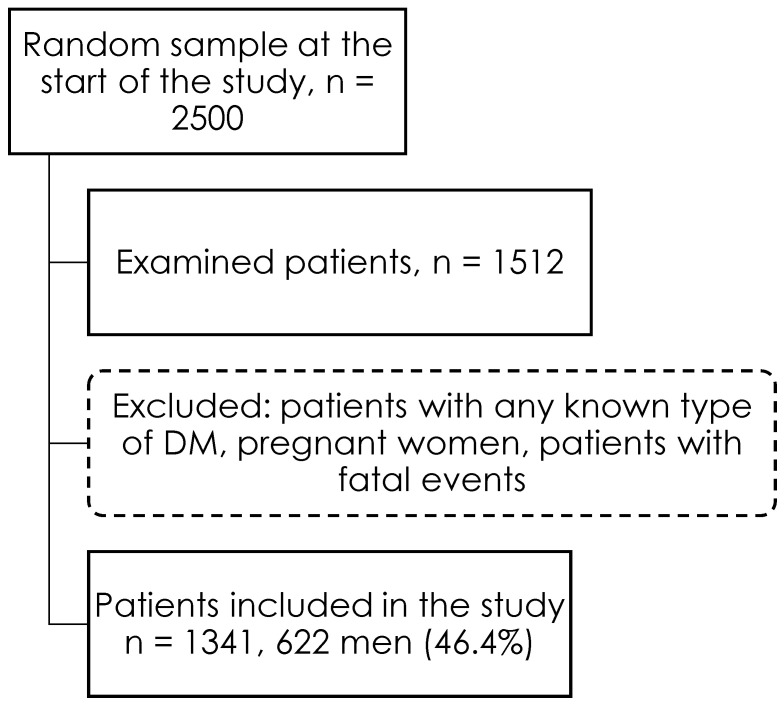
Flow chart of the study population according to inclusion and exclusion criteria.

**Table 1 jpm-13-01403-t001:** Characteristics of the studied sample of 25–44-year-old participants in Novosibirsk.

Parameters	Patients with Developed T2DM *n* = 11	Patients without T2DM *n* = 1330	*p*
Age	39 (31; 44)	37 (31; 41)	0.253
Men (*n*, %)	6 (54.5%)	616 (46.3%)	0.763
Women (*n*, %)	5 (45.5%)	714 (53.7%)	
Physical activity less than 3 h/week (*n*, %)	9 (81.8%)	882 (66.6%)	0.355
Smoking (*n*, %)	5 (45.5%)	438 (33.1%)	0.386
Married (*n*, %)	7 (63.6%)	953 (71.9%)	0.515
Employed (*n*, %)	7 (63.6%)	1108 (83.6%)	0.076
Higher education (*n*, %)	2 (18.2%)	833 (62.9%)	0.003
WC, sm	101.00 (90.00; 122.00)	85.00 (76.0; 95,40)	<0.001
SBP, mm Hg	132.50 (121.50; 140.00)	119.00 (110.00; 129.00)	0.033
DBP, mm Hg	84.50 (76.00; 96.00)	78.00 (71.00; 86.00)	0.095
Presence of AH	4 (36.4%)	237 (17.9%)	0.120
BMI, kg/m^2^	32.37 (29.78; 39.44)	25.01 (22.04; 28.70)	<0.001
≤25 kg/m^2^	0	663 (49.9%)	
25–29.9 kg/m^2^	3 (27.3%)	425 (32.0%)	<0.001
≥30 kg/m^2^	8 (72.7%)	240 (18.1%)	<0.001
TG, mmol/L	1.75 (1.22; 2.89)	0.94 (0.68; 1.38)	0.003
TG ≥ 1.7 mmol/L (*n*, %)	7 (63.6%)	216 (16.3%)	0.001
HDL-C, mmol/L	1.03 (10.90; 1.16)	1.29 (1.08; 1.52)	0.001
HDL < 1 mmol/L for men and <1.2 mmol/L for women (*n*, %)	7 (63.6%)	389 (29.4%)	0.020
LDL-C, mmol/L	2.85 (2.17; 3.77)	3.13 (2.53; 3.69)	0.370
LDL-C ≥ 3 mmol/L (*n*, %)	4 (36.4%)	739 (55.9%)	0.231
TCH, mmol/L	5.22 (3.85; 5.68)	4.96 (4.34; 5.63)	0.715
TCH ≥ 5 mmol/L (*n*, %)	6 (54.5%)	646 (48.9%)	0.769
Creatinine, umol/L	72.00 (65.00; 81.50)	74.00 (67.00; 82.00)	0.701
GFR _CKD-EPI_, mL/min/1.73 m^2^	1 (10.0%)	220 (23.6%)	0.467
GFR _CKD-EPI_ < 90 mL/min/1.73 m^2^ (*n*, %)	106.38 (94.98; 110.323)	101.47 (90.66; 110.21)	0.505
Albumin, g/L	41.65 (38.65; 43.93)	42.70 (40.80; 44.60)	0.267
Urea, mmol/L	3.95 (3.05; 5.63)	4.30 (3.70; 5.20)	0.534
Glucose, mmol/L	5.83 (5.31; 6.98)	5.73 (5.31; 6.04)	0.251
Glucose ≥ 6.1, mmol/L	4 (36.4%)	292 (22.1%)	0.257

Note: AH—arterial hypertension, BMI—body mass index, DBP—diastolic blood pressure, HDL-C—high-density lipoprotein cholesterol, LDL-C–low-density lipoprotein) cholesterol, SBP—systolic blood pressure, TCH—total cholesterol, TG—triglycerides, WC—waist circumference.

**Table 2 jpm-13-01403-t002:** One-factor logistic regression analysis of the association of cardiometabolic risk factors with the development of diabetes mellitus adjusted for gender and age.

Indicators	Logistic Regression Analysis			
	OR	95% Confidence Interval (CI)		*p*
		Lower Bound	Upper Bound	
BMI, per 1 kg/m^2^	1.200	1.119	1.287	<0.001
SBP, per 10 mm Hg	1.357	1.010	1.842	0.050
TG, per 1 mmol/L	1.475	1.079	2.017	0.015
TG ≥ 1.7 mmol/L vs. TG < 1.7 mmol/L	9.013	2.491	32.614	0.001
HDL-C, per 1 mmol/L	0.022	0.002	0.293	0.004
HDL-C < 1 mmol/L for men (vs. ≥1 mmol/L) and <1.2 mmol/L for women (vs. ≥1.2 mmol/L)	4.413	1.271	15.322	0.019
WC, by 1 cm	1.086	1.049	1.123	<0.001
AO (WC ≥ 80 cm vs. WC < 80 cm for women and ≥94 cm vs. WC < 94 cm for men)	12.967	1.634	102.927	0.015
Level of education, higher education vs. other types of education	7.014	1.484	33.151	0.014

Note: AO—abdominal obesity, BMI—body mass index, HDL-C—high-density lipoprotein cholesterol, SBP—systolic blood pressure, TG—triglycerides, WC—waist circumference.

**Table 3 jpm-13-01403-t003:** Multivariate logistic regression analysis of the association of cardiometabolic risk factors with the development of diabetes mellitus.

Analyzed Factors	Model 1		Model 2		Model 3		Model 4	
	OR (95% CI)	*p*	OR (95% CI)	*p*	OR (95% CI)	*p*	OR (95% CI)	*p*
Gender, male vs. female	1.083 (0.308–3.811)	0.901	0.992 (0.265–3.711)	0.990	0.884 (0.226–3.454)	0.860	1.353 (0.367–4.986)	0.650
Age, per 1 year	1.029 (0.922–1.149)	0.606	1.053 (0.938–1.183)	0.381	1.043 (0.925–1.176)	0.493	1.018 (0.912–1.137)	0.747
TG ≥ 1.7 mmol/L vs. TG < 1.7 mmol/L	5.314 (1.424–19.833)	0.013	5.119 (1.345–19.477)	0.017	5.365 (1.371–20.995)	0.016	5.220 (1.348–20.217)	0.017
Availability of AO vs. absence of AO	7.893 (0.945–65.914)	0.056	-	-	-	-	-	-
BMI, per 1 kg/m^2^	-	-	1.184 (1.098–1.277)	<0.001	1.183 (1.083–1.293)	<0.001	-	-
BMI ≥ 30 kg/m^2^ vs. BMI < 30 kg/m^2^	-	-	-	-	-	-	6.461 (1.600–26.098)	0.009
Level of education, all types of education except higher vs. higher education	-	-	-	-	5.172 (1.027–26.035)	0.046	5.649 (1.178–27.094)	0.030

Note: AO—abdominal obesity, BMI—body mass index, TG—triglycerides, WC—waist circumference.

## Data Availability

Not applicable.

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
