# Peer review of "Triglycerides, Obesity and Education Status Are Associated with the Risk of Developing Type 2 Diabetes in Young Adults, Cohort Study"

_jpm, 2023, doi:10.3390/jpm13091403_

Round 1
Reviewer 1 Report
General comments:
The manuscript entitled “Triglycerides, obesity and education status are associated with the risk of developing type 2 diabetes in young people, cohort study” by Garbuzova E.V. et al, is in the scope of the journal and presents an interesting original article that addresses a topic with an interest for the biomedical and medical community.
Overall, this is an interesting study. The literature search is adequate. The performed analyses are appropriate, the results are well presented, the findings are promising and the conclusions are supported by the data. Nonetheless, some points should be amended:
Specific Comments:
1. Figure 1 should be of better quality.
2. There are typos and errors in the aspect of English writing. Please check throughout the text to correct them.
3. Please change “body weight” to “body mass” throughout the manuscript.
4. All abbreviations need to be explained at first use in the text and Abstract and Figure legends, and when introduced the abbreviation should be exclusively used throughout the whole text (e.g. DMT2 in line 44, Tg in line 137….).
5. What are the future possibilities or research directions that could be undertaken as a result of the findings of this study?
There are typos and errors in the aspect of English writing. Please check throughout the text to correct them.
Author Response
Dear reviewer,
Thank you for your appreciation of our work! We tried to take into account all the comments.
- We tried to make a picture of the best quality.
- The text has been verified by a professional medical translator.
- We have changed “body weight” to “body mass” throughout the manuscript.
- Thank you for the comment. We have explained all the abbreviations in the text.
- The data can be used to study T2DM risk factors in younger populations in other regions. Also these data can be taken into account when developing personalized preventive measures for different age groups.
Reviewer 2 Report
Dear authors,
I have read your paper with interest. I find it exciting and concisely written. Yet, I have some issues to be addressed.
- Throughout the manuscript, I think you should use the widely established abbreviation for Type 2 Diabetes Mellitus T2DM (this is the most accepted abbreviation in most medical manuscripts).
- In the Abstract, you write that the patients were enrolled from 2013-2017 and that you have identified cases until 2023. This can cause confusion that the study could be a retrospective one, when in fact it is a prospective one.
- The second phrase from the introduction is poorly written, one cannot understand clearly which percentage goes where.
- Furthermore, if there is so clear data about T2DM in young Russian adults, what is the relevance of your study?
- I think that the authors could emphasize even from the title that the study addresses the Russian population from the Siberian town of Novosibirsk, because this could be used as an argument for originality.
- At row 173 you say that low BMI is associated with the risk of developing T2DM.
- You have discussed the limitations, but not the strengths and innovations used in the creation/conduction of your study.
- The ”young people” term can lead to disambiguation, I would prefer more „young adults”.
- I feel the Discussion section can be enriched.
Good luck!
Dear authors,
I feel there is room for English language improvement in your manuscript. Some phrases can lead to ambiguous meanings and can be rephrased.
Good luck!
Author Response
Dear reviewer,
Thank you for your appreciation of our work! We tried to take into account all the comments.
- We have changed the abbreviation for Type 2 Diabetes Mellitus (T2DM) throughout the text.
- We tried to explain in the annotation that the collection of endpoints was carried out later, from 2019 to 2023.
- We have rephrased this sentence.
- The relevance of the study lies in the fact that the main risk factors that could affect the development of T2DM in young people were studied. We did not aim to study the prevalence of T2DM in young people, and its description was only one of the first stages of the work.
- The studied population is described in detail in the materials and methods section; the indication of the region and city in the title of the work remains at the discretion of the authors.
- Row 173. Thank you for your care. This is a typo; we have fixed it.
- We have added this information to the article.
- We have changed ”young people” to „young adults”.
- We have tried to enrich the discussion section due to the possible pathogenetic connection of the studied conditions.
- The text has been verified by a professional medical translator.